# Relevance and Scale Dependence of Hydrological Changes in Glacierized Catchments: Insights from Historical Data Series in the Eastern Italian Alps

**Luca Carturan [1,2,*]**, **Fabrizio De Blasi [2]**, **Federico Cazorzi [3]**, **Davide Zoccatelli [2]**,
**Paola Bonato [2]**, **Marco Borga [2]** and **Giancarlo Dalla Fontana [2]**

1   Department of Geosciences, University of Padova, Via Gradenigo, 6, 35131 Padova, Italy
2   Department of Land, Environment, Agriculture and Forestry, University of Padova, Viale dell'Università, 16,
    35020 Legnaro (Padova), Italy; fabrizio.deblasi@unipd.it (F.D.B.); zoccatelli.davide@gmail.com (D.Z.);
    paola.bonato.1@gmail.com (P.B.); marco.borga@unipd.it (M.B.); giancarlo.dallafontana@unipd.it (G.D.F.)
3   Department of Agricultural, Food, Environmental and Animal Sciences, University of Udine,
    Via delle Scienze, 206, 33100 Udine, Italy; federico.cazorzi@uniud.it
*   Correspondence: luca.carturan@unipd.it

**Abstract:** Glaciers have an important hydrological buffering effect, but their current rapid reduction raises concerns about future water availability and management. This work presents a hydrological sensitivity analysis to different climatic and glacier cover conditions, carried out over four catchments with area between 8 and 1050 km$^2$, and with glacierization between 2% and 70%, in the Italian Alps. The analysis is based on past observations, and exploits a unique dataset of glacier change and hydro-meteorological data. The working approach is aimed at avoiding uncertainties typical of future runoff projections in glacierized catchments. The results highlight a transition from glacial to nival hydrological regime, with the highest impacts in August runoff over smaller catchments. The buffering effect of current glaciers has largely decreased if compared to the Little Ice Age, up to 75% for larger catchments, but it is still important during warm and dry summers like 2003. We confirm a non-linear relationship between glacier contribution in late summer and catchment area/percent glacierization. The peak in runoff attributable to glacier melt, expected in the next 2–3 decades on highly glacierized alpine catchments, has already passed in the study area.

**Keywords:** glacier runoff change; climate change; alpine watersheds; Italian Alps; glacio-hydrological modeling; historical data series

## 1. Introduction

High altitude mountain environments are particularly sensitive to climate change, and the European Alps are one of the mountain regions of the world where effects from atmospheric warming are more visible. In particular, the Alpine cryosphere is experiencing very rapid and intense transformations, the most dramatic of which is glacier shrinking and downwasting [1–3].

Knowledge of the hydrological response of high-altitude watersheds is critical to manage water resources under current climate change, which is characterized by higher temperatures, lower percentage of solid precipitation, temporal redistribution and quantitative variations in precipitation inputs, and more persistent drought conditions in summer [4].

In many catchments of the Alps and neighboring lowlands, glaciers have a significant hydrological buffering effect, securing water supply during warm and dry summer periods due to increased meltwater runoff. This effect was particularly clear in recent warm years, when glacier runoff contribution was remarkable not only in headwater catchments, but also on downstream macroscale

watersheds originating in the Alps, like the Po and Adige in Italy, that have a small percent glacier cover [5]. For these reasons and due to the concern about impacts of sustained climatic forcing on water resources, recent investigations have focused on the current and future role of glaciers in the hydrological response of Alpine catchments e.g., [6–8].

The results from these studies highlight significant changes in the future hydrology of glacierized alpine catchments. Besides a ubiquitous shift from glacial to nival-glacial or nival runoff regime, there is a case-specific response for individual catchments, depending on the timing of the runoff peak induced by glacier wasting, and on the scale dependence of hydrological changes, which is related to the importance of glacier contribution to runoff and which does not scale linearly with the percentage of glacierization [5,9]. Glacier wasting can lead to an increase in overall streamflow if the rate of volume loss is sufficiently large, or to a decline in streamflow if the glacierized area decreases at a rate which is sufficient to offset any increase in the ablation rate [10]. Several studies have demonstrated the in many regions the peak in runoff has already passed e.g., [11,12].

In absence of high-quality long observation series of glacier change and streamflow, which would enable a direct quantification of these changes, these points require the application of coupled glacio-hydrological and glacier dynamic models, normally calibrated and validated with recent measurements and run to project future conditions employing simulated climatic scenarios. Models with variable degrees of sophistication have been developed, from simple lumped models based on the degree-day algorithm, to fully-distributed physically based approaches. All of them have to deal with the high complexity and climatic sensitivity of glacio-hydrological systems, combined with the large uncertainty of future modeled climatic scenarios [13,14]. Model generalizability depends also on the quality of meteorological input data and their spatialization [15,16], and on the availability of data for internal validation.

Here we present a sensitivity analysis based on a comprehensive dataset of glacier change observations and reconstructions, and on long time series of hydro-meteorological data from the Noce River Basin in the Italian Alps. Some of these records are amongst the longest available at high elevation in the Alps. The analysis is intended to provide (i) an assessment of hydrological changes in response to glacier shrinking and (ii) an evaluation of the scale effects on the hydrological changes in glacierized catchments. The working approach consists in the application of a modeling tool for analyzing the hydrological behavior of four catchments with different sizes, under different (observed) climatic conditions, and with varying glacier coverage. The peculiarity of this study lies in the use of historical data, which enabled minimizing modeled variables and working with a high degree of internal model consistency, constrained by actual observations.

## 2. Study Area

The Val di Sole is located in the upper part of the Noce River catchment, a right-hand tributary of the Adige River, which is the main river system in northeastern Italy. The elevation in this 1046 km$^2$ wide catchment ranges between 380 m a.s.l. at the outlet (Tassullo) and 3769 m a.s.l. at the summit of Mt. Cevedale, averaging 1705 m a.s.l (Figure 1). Metamorphic rocks (mica schists, paragneiss and orthogneiss) prevail north of the main longitudinal axis of the valley, corresponding with the Noce River. This area (72% of the total) belongs to the Ortles-Cevedale Group, the largest glacierized mountain group in the Italian Alps. Tonalite is found in the south-western part, which belongs to the Adamello-Presanella mountain group (9% of the total), whereas dolomites and limestones are prevalent in the south eastern part (19% of the study area, www.protezionecivile.tn.it).

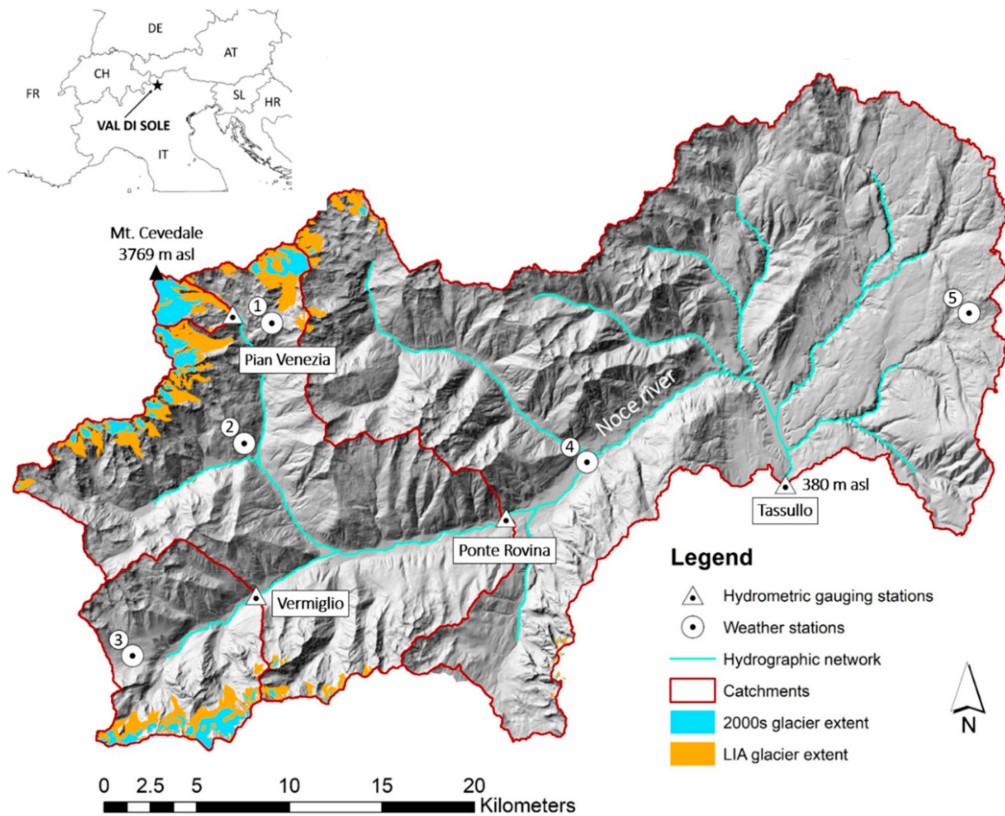

**Figure 1.** Geographical setting of the study area. Weather stations: (1) Careser Diga, (2) Peio, (3) Tonale, (4) Malè, (5) Mendola.

The catchment includes a glacierized area (16 km$^2$ in 2006), although the glacier cover is strongly reduced compared to the Little Ice Age (LIA maximum 45 km$^2$ between the 16th and 19th centuries, Zanoner et al. [17]). Bare rocks and debris are found outside the glaciers down to an elevation of 2700 m. Between 2200 m and 2700 m there is a discontinuous cover of alpine meadows and shrubs, while below 2200 m, there is an almost continuous forest coverage, with the exception of the valley flow under 1000 m which is occupied by cultivations and settlements (www.protezionecivile.tn.it).

This area was selected because of the good availability of hydro-meteorological data, to some extent attributable to the hydropower development (two power plants operate close to Pian Venezia and Peio, with a total of 70 MW power installed). Measurement series date back to the 1920s, and glacier change observations begun at the end of the 19th Century [18–22]. High altitude weather stations and hydrometric stations started to operate between the 1920s and 1930s. Based on the availability of stream flow data, we were able to perform our analyses in four catchments with areas ranging from 8.4 to 1046 km$^2$ and present percent glacierization from 2 to 45%. Figure 1 shows these catchments and their principal characteristics are summarized in Table 1.

**Table 1.** Topographic characteristics of the four investigated catchments. LIA glacierized area from Zanoner et al. [17]; 2000s glacierized area from Salvatore et al. [23].

| Catchment | Area (km$^2$) | Mean Elevation (m) | Minimum Elevation (m) | Maximum Elevation (m) | LIA Glacierized Area | 2000s Glacierized Area |
|---|---|---|---|---|---|---|
| Tassullo | 1046.1 | 1705 | 380 | 3759 | 4% | 2% |
| Ponte Rovina | 385.6 | 2148 | 783 | 3759 | 11% | 4% |
| Vermiglio | 79.8 | 2210 | 1173 | 3556 | 14% | 6% |
| Pian Venezia | 8.4 | 3082 | 2295 | 3759 | 69% | 45% |

## 3. Data Collection and Processing

### 3.1. Hydro-Meteorological and Glacier Mass Balance Data

Daily temperature and precipitation data from five weather stations were used in this study. The weather stations are rather well spatially distributed in the study area (Figure 1) and cover an elevation range between 730 and 2607 m. These stations have been managed by the Ufficio Idrografico of the Magistrato alle Acque di Venezia until the 1990s, and they are now managed by the Provincia Autonoma di Trento. The data series are almost continuous since the 1920s, with few exceptions (Figure 2).

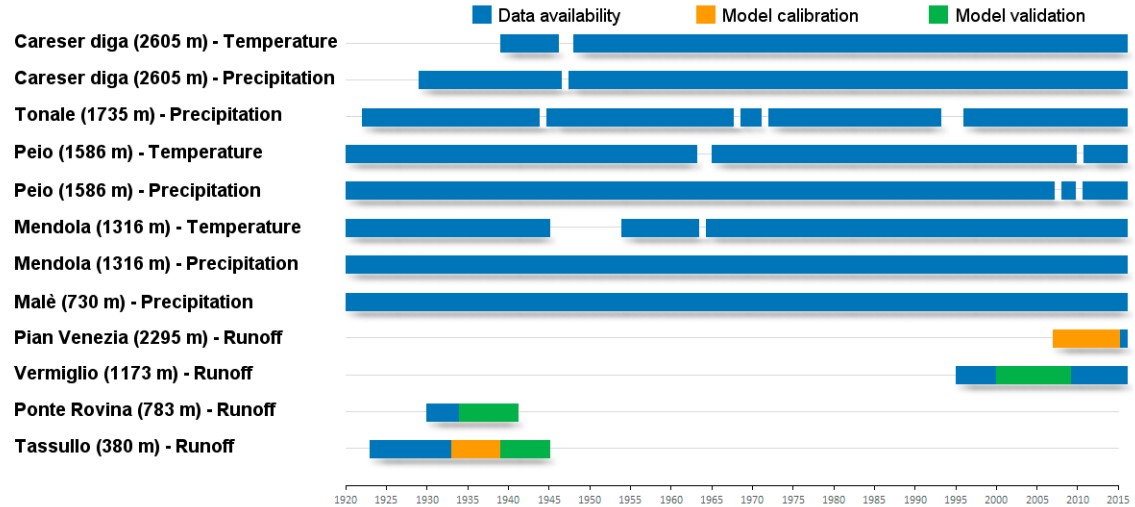

**Figure 2.** Availability of hydro-meteorological data in the study area.

The data and metadata were inspected to identify potential measurement problems arising from station moves, changes in instrumentation, changes in surrounding environmental characteristics and structures, and observation practices. These observations have been helpful for identifying inhomogeneities in the meteorological series, that were homogenized according to guidelines and statistical procedures prescribed in literature [24,25]. After homogenization, the series were gap-filled using linear regressions between different stations for temperature, and ratios of monthly precipitation data. The precipitation data were also corrected for gauge under catching using the procedure implemented by Carturan et al. [26] for the weather stations in the study area.

Data series from four hydrometric stations were useable in this work, located at the outlet of catchments with a significant range of area, elevation and percentage of glacierization (Table 1). The gauging stations of the two larger catchments, Tassullo and Ponte Rovina, have data from the 1920s to the 1940s, whereas the Vermiglio and Pian Venezia stations have data from the last two decades.

Glacier mass balance observations were used as model constrains in this study. They consist of point mass balance observations collected on the La Mare Glacier [27], starting in 2003. Point observations were carried out each year at two-to-four-week intervals from May to September, and the average number of measuring points on the glacier was 37. Observations consist of snow accumulation measurements and snow, firn and ice ablation measurements, collected by means of standard procedures in use for the 'glaciological' method [28]. Point observations were complemented by detailed snow cover maps, derived from manual digitization of oblique terrestrial photos [29]. Snow cover maps of lower detail were also available for the entire study area, derived from automatic classification of Landsat TM and MODIS imagery (courtesy of Eurac Research, [30,31]).

*3.2. Glacier Extent and Topography*

The sensitivity analysis was carried out using glacier extents during the Little Ice Age (LIA), in the 2000s (Current), and a condition of complete deglaciation (Noglac). These three conditions were selected based on available topographic data and glacier extent reconstructions. In particular, the recent work from Zanoner et al. [17] provided the areal extent of all the glaciers in the study area during the LIA maximum, reconstructed according to documentary and geomorphological evidence and mapped with the aid of a 2-m resolution Lidar DTM surveyed in 2006. The Current glacier extent was mapped by Salvatore et al. [23], using high-resolution orthophotos acquired in 2006, at the same time as the DTM.

In the area outside the glaciers, and for the glacierized area in 2006, the surface topography was derived from the 2006 2-m DTM. For the glacierized area in the LIA, and for the bedrock underneath the 2006 glaciers (to be used for the Noglac scenario), we had to calculate the surface topography. This was accomplished in two steps: (i) calculation of the bedrock topography using the method proposed by Huss and Farinotti [32], and (ii) calculation of the surface topography of the LIA, from the bedrock topography, exploiting the method proposed by Benn and Hulton [33].

The Huss and Farinotti [32] method calculates the spatially distributed thickness of glaciers starting from the surface topography, using a physically based approach and a simple dynamic model. We applied the method to glacier surface topographies extracted from a DTM surveyed in 1983, which was a period when glaciers were much closer to dynamic equilibrium than in 2000s. We recalibrated the parameters in agreement with mass balance gradients and geophysical data retrieved in the study area. Validation was carried out using geophysical data independent from calibration and ice thickness obtained differencing the 2006 and 1983 DTMs in deglaciated terrain. The bedrock topography was finally obtained subtracting the calculated ice thickness from the glacier surface DTM, and merged with the 2006 DTM in the areas outside the glaciers, to obtain the catchment DTM for the Noglac scenario.

The Benn and Hulton [33] method enabled the calculation of the LIA glacier surface elevation along longitudinal profiles, starting from the bedrock topography calculated in the previous step and from the front positions reported by Zanoner et al. [17]. The method uses an exact solution of a 'perfectly plastic' glacier model, implemented in an ExcelTM spreadsheet. According to the authors' suggestions, we used target ice elevations derived from geomorphological mapping and let the yield stress vary along the flowlines, incorporating the effect of valley-side drag on the glacier profile. The method was independently validated using old topographic maps surveyed at the beginning of the 20th Century on some glaciers [34,35].

The surface topography of the LIA glaciers was then reconstructed drawing manually the elevation contours at 50 m intervals. We started each contour from the LiDAR 2006 elevation along the LIA glaciers' margin, using the longitudinal profiles obtained with the Benn and Hulton [33] method as constrains in the middle part of the glaciers. Here, the degree of subjectivity would be highest in absence of constrains, in particular across the accumulation area. The surface topography obtained in this manner was merged with the 2006 DTM in the areas outside the LIA glaciers, to obtain the catchment DTM for the LIA glacierization scenario. All DTMs were resampled to a resolution of 30 m, which is a good compromise between spatial detail and model computation efficiency.

Further details on the application of these methods can be found in De Blasi [36].

## 4. Sensitivity Analysis

*4.1. Working Approach*

The sensitivity analysis was carried out comparing three decades that had highly different mean meteorological conditions ('meteorological periods' hereafter), selected also for the occurrence of temperature and precipitation extremes (Table 2). After being carefully calibrated and validated with available observations, a glacio-hydrological model was used for the sensitivity analyses, running it over the three different decades. We chose to focus on 10-year periods because (i) glacier size can

be assumed time-invariant, (ii) there is a good smoothing of single-year extreme conditions, while (iii) preserving a good characterization for climatic periods with distinct characteristics.

**Table 2.** Air temperature and precipitation anomalies for the meteorological periods and single extreme years analyzed in this study (reference period: 1961–2010).

| | | Meteorological Period | | | Extreme Years | |
|---|---|---|---|---|---|---|
| | | 1940s (1941–1950) | 1970s (1968–1977) | 2000s (2003–2012) | 1967–1968 | 2002–2003 |
| **Precipitation** | Winter anomaly (Oct–Apr) | −25% | −2% | −4% | −18% | −10% |
| | Summer anomaly (May–Sep) | −10% | +11% | −8% | +32% | −32% |
| **Temperature** | Winter anomaly (Oct–Apr) | −0.2 °C | −0.4 °C | +0.4 °C | −0.3 °C | +0.4 °C |
| | Summer anomaly (May–Sep) | +0.8 °C | −0.8 °C | +1.1 °C | −1.3 °C | +3.3 °C |

All the possible combinations of three meteorological periods, four catchments (described in Section 2) and three glacier extents (described in Section 3.2) have been tested, obtaining 36 model runs (Table 3). The model provided in output continuous daily time series of runoff, which were analyzed and compared. In particular, we investigated the differences in the hydrologic response at the daily, monthly and seasonal timescales, considering decadal averages but also analyzing extreme single-year case studies in 2003 and 1968 (Table 2).

**Table 3.** Combinations of meteorological periods, catchments, and glacier extents tested in the sensitivity analyses.

| | 1940s | | | 1970s | | | 2000s | |
|---|---|---|---|---|---|---|---|---|
| (1) | Tassullo | LIA | (13) | Tassullo | LIA | (25) | Tassullo | LIA |
| (2) | Tassullo | Current | (14) | Tassullo | Current | (26) | Tassullo | Current |
| (3) | Tassullo | Noglac | (15) | Tassullo | Noglac | (27) | Tassullo | Noglac |
| (4) | Ponte Rovina | LIA | (16) | Ponte Rovina | LIA | (28) | Ponte Rovina | LIA |
| (5) | Ponte Rovina | Current | (17) | Ponte Rovina | Current | (29) | Ponte Rovina | Current |
| (6) | Ponte Rovina | Noglac | (18) | Ponte Rovina | Noglac | (30) | Ponte Rovina | Noglac |
| (7) | Vermiglio | LIA | (19) | Vermiglio | LIA | (31) | Vermiglio | LIA |
| (8) | Vermiglio | Current | (20) | Vermiglio | Current | (32) | Vermiglio | Current |
| (9) | Vermiglio | Noglac | (21) | Vermiglio | Noglac | (33) | Vermiglio | Noglac |
| (10) | Pian Venezia | LIA | (22) | Pian Venezia | LIA | (34) | Pian Venezia | LIA |
| (11) | Pian Venezia | Current | (23) | Pian Venezia | Current | (35) | Pian Venezia | Current |
| (12) | Pian Venezia | Noglac | (24) | Pian Venezia | Noglac | (36) | Pian Venezia | Noglac |

Meteorological period
Catchment
Glacier extent

## 4.2. The Glacio-Hydrological Model

The model used in this work is EISModel 4.0, which has been developed in various steps in the last 20 years, and was optimized for applications in glacierized catchments [26,36]. It consists of two modules, the first that processes the input data and calculates the mass balance of the snowpack and glaciers, and the second that is used for runoff routing (Figure 3).

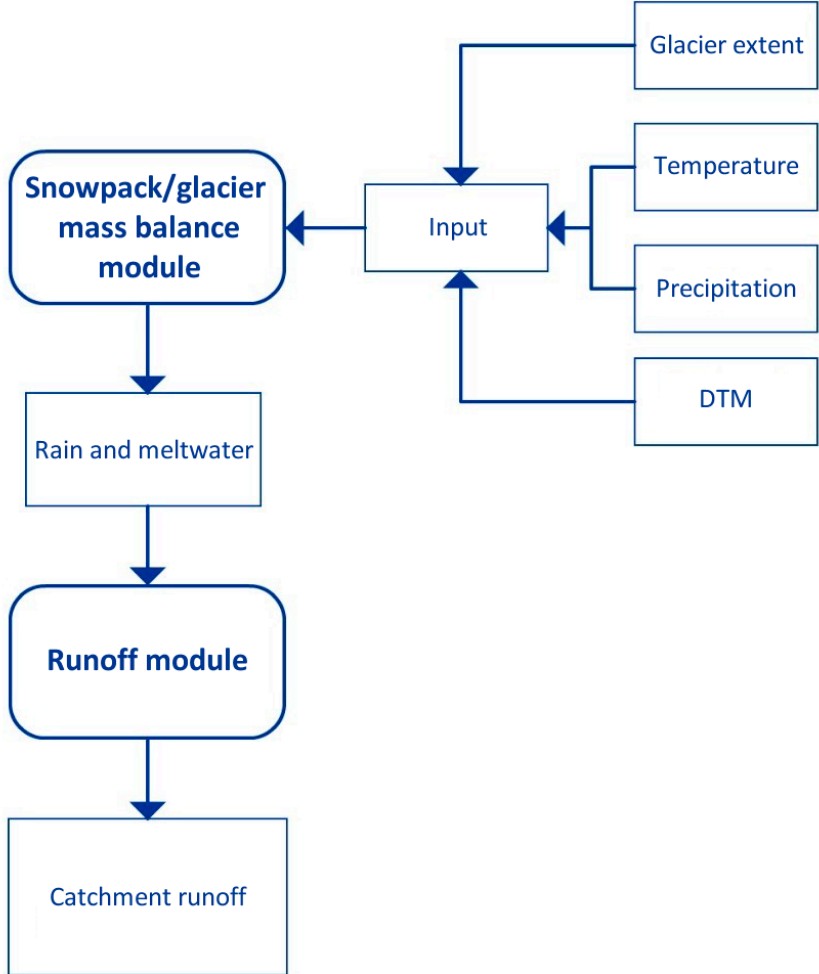

**Figure 3.** Simplified scheme of the EISModel 4.0 structure.

The first module is a fully-distributed model that belongs to the so-called enhanced temperature index (ETI) melt models category. It is derived from the model described by Cazorzi and Dalla Fontana [37], and simulates accumulation and melt processes requiring as inputs the DTM of the watershed and meteorological data (precipitation and 2 m air temperature) from at least one weather station. Here we describe the main algorithm and adaptations for the present work, referring the reader to Carturan et al. [26] for a detailed description of the model.

The model simulates snow accumulation adding a new layer at each snowfall, accounting for wind and avalanche redistribution by means of a snow redistribution factor computed off-line at the pixel scale. The three main melt algorithms described in literature for ETI models can be used alternatively to simulate melt processes. In this work, we used the Cazorzi and Dalla Fontana [37] melt algorithm

$$\mathbf{MLT_{X,t} = \ CMF \cdot TM_{X,t} \cdot CSR_{X,t}(1 - \alpha_{X,t})} \tag{1}$$

where $MLT_{X,t}$ is the melt rate (mm d$^{-1}$) at the pixel X at time t, $TM_{X,t}$ (°C) is air temperature, $\alpha_{X,t}$ is the surface albedo, $CSR_{X,t}$ (W m$^{-2}$) is the clear sky radiation and CMF (mm d$^{-1}$ °C$^{-1}$ W$^{-1}$ m$^2$) is a combined melt factor that is the main calibration factor of the model.

The model required some modifications for this study, namely the transition from hourly to daily time steps, which is necessary to manage historical data series, and improvements in the spatialization of meteorological inputs over large catchments (the model was mainly applied to small catchments at high altitude so far). Two different temperature lapse rates were used above/below a threshold elevation, which separates the upper areas of the catchment from the valley floor, where strong thermal

inversions occur. The threshold elevation for thermal inversions, identified plotting air temperature data versus elevation, corresponds to the Peio weather station (1586 m), whose temperature data were used for extrapolations at higher/lower altitude. To account for seasonal effects, we also used mean monthly lapse rates. All lapse rates and the threshold elevation were calculated and averaged from the historical data series described in Section 3.1, and are shown in Figure 4.

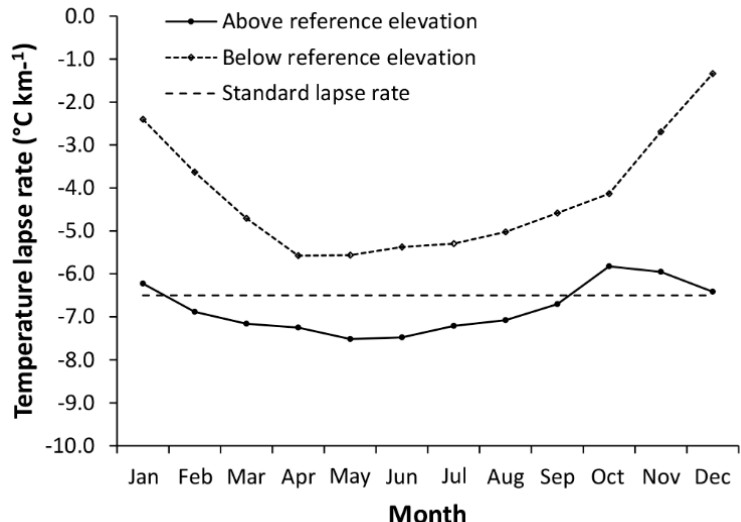

**Figure 4.** Mean monthly temperature lapse rates applied above/below the reference elevation (Peio weather station, 1586 m) for thermal inversions.

A similar approach was used to compute monthly variable mean vertical precipitation gradients. To account for the horizontal variability in precipitation, the raw data were normalized to the mean elevation of the catchment (by means of the precipitation gradients), spatialized using inverse distance weighting (IDW), and then extrapolated at each pixel's elevation re-applying the vertical precipitation gradients.

The classification of precipitation as rain or snow was optimized using direct observations at the Careser Diga weather station. We introduced a buffer of 1 °C above/below the threshold temperature (2.2 °C), and used a fractional increase from 100% solid at 1.2 °C, to 50% solid at 2.2 °C, to 100% liquid at 3.2 °C. This solution was required for model applications at daily time steps, because a simple switch from 100% solid to 100% liquid at the threshold temperature, as in use for the hourly version, did not enable a phase partitioning in agreement with direct observations.

A further improvement was the introduction of a cloud correction factor (CCF), to account for the diurnal cycle of cloud cover and its effects in the spatial distribution of the global radiation. This recursive cloud formation cycle, triggered by thermal convection, is often recognizable in many alpine areas during the warm season [38,39]. The $CSR_{X,t}$ in Equation (1) is therefore adjusted to $RCSR_{X,t}$ as

$$RCSR_{X,t} = CSR_{X,t} \left(1 - CCF\ CI_{X,J}\right) \tag{2}$$

where $CI_{X,J}$ is a cloud index calculated in function of the local aspect, varying between 0 at sunrise azimuth and 1 at sunset azimuth.

A semi-lumped module based on the well-known Stanford watershed model [40] was used for runoff routing of soil surface water, generated in output from the first module as meltwater at the bottom of the snowpack/glaciers and rain on bare ground. The routing module performs calculations following the classical scheme: evapotranspiration—infiltration—upper zone processes—lower zone processes. The catchment was subdivided into three elevation bands, which were identified based on vegetation and soil use characteristics. For each band, we used different parameters to account for their specific hydrological response. The upper band comprises the catchment area above 2200 m, which is

above the treeline and has mainly impervious or thin soil. The middle band is located between 1000 and 2200 m, and is mainly covered by coniferous forest with more developed and thick soils. The lower band, below 1000 m, is mainly occupied by alluvial deposits with grassland, crops, and small villages.

Most morphological, vegetation, and hydrological input parameters were derived from the DTM and from a detailed land-use map available for the study area. The optimization of the calibration parameters UZSN (nominal upper zone storage), LZSN (nominal lower zone storage), CB (infiltration index), CC (volume of the subsurface outflow), and ETPFRC (reduction factor of the potential evapotranspiration) is described in the following section.

### 4.3. Model Calibration and Validation

The good availability of experimental data enabled a separate calibration of the two modules of EISModel 4.0, and internal validation. We applied a hierarchical testing approach in calibration and validation, aimed at maximizing model transferability in space (different catchments and glacierization) and time (different meteorological periods), and inspired by the works of Klemes [41], Hanzer et al. [42], and Santos et al. [43].

The snowpack-glacier module was calibrated and validated using mass balance measurements available for the La Mare Glacier and snow cover maps. Calibration and validation assessments were based on the Nash and Sutcliffe [44] Index (N&S) for point balances, and on confusion matrices for snow cover maps. Independent validation test were carried out on mass balance points/years/snowmap dates and areas different from those used in calibration. Both calibration and validation provided robust results, with N&S above 0.90 for point balances, correct snow cover classification above 80%, and N&S above 0.65 for runoff, highlighting good model transferability (Table 4).

**Table 4.** Results of EISModel 4.0 calibration and independent validation tests. KGE is the Gupta et al.'s [45] criterion.

| Variable | Calibration Statistics | | | Validation Statistics | | |
|---|---|---|---|---|---|---|
| Point mass balance | N&S | $r^2$ | ME (mm $y^{-1}$) | N&S | $r^2$ | ME (mm $y^{-1}$) |
| | 0.984 | 0.984 | +12 | 0.927 | 0.933 | +97 |
| | **Calibration Statistics** | | | **Validation Statistics** | | |
| Snow cover classification | % correct classification | % too early melt | % too late melt | % correct classification | % too early melt | % too late melt |
| | 85.6 | 8.6 | 5.9 | 82.3 | 11.7 | 6.0 |
| | **Calibration Statistics** | | | **Validation Statistics** | | |
| Runoff | N&S | KGE | ME (%) | N&S | KGE | ME (%) |
| | 0.800 | 0.887 | −1.1 | 0.665 | 0.750 | +3 |

The runoff routing module was calibrated and validated using streamflow data available on catchments described in Section 2. The module was run testing different combinations of parameters, keeping unchanged those optimized for the snowpack/glacier module. Runoff routing parameters were tested within physically-meaningful ranges, indicated by the guidelines from Crawford and Linsley [40]. We first calibrated the parameters of the upper elevation band using data from the Pian Venezia catchment. The parameters for the middle and lower elevation bands were then calibrated using the Tassullo catchment, keeping unchanged the calibration of the upper band. Independent validations were carried out in the period and catchments indicated in Figure 2. The model worked well in calibration and its performance was good also in validation (Table 4).

## 5. Results

### 5.1. Runoff Regime under Different Meteorological Periods and Glacier Extent

In the Pian Venezia catchment the annual runoff peak is in July or August, with all glacier extents (Figure 5). For all meteorological periods, a shift towards earlier and lower runoff peaks is observable

moving from the LIA and Current glacier extents to the Noglac condition. The percent difference among glacier cover conditions increases throughout the summer, reaching a maximum between August and October, suggesting that in these periods the runoff is mainly fed by glacier melt. Late summer differences among glacier extents are smaller for the 1970s, compared to the warmer 1940s and 2000s, when the Current glacier contribution is calculated about halfway between the LIA and the Noglac extents.

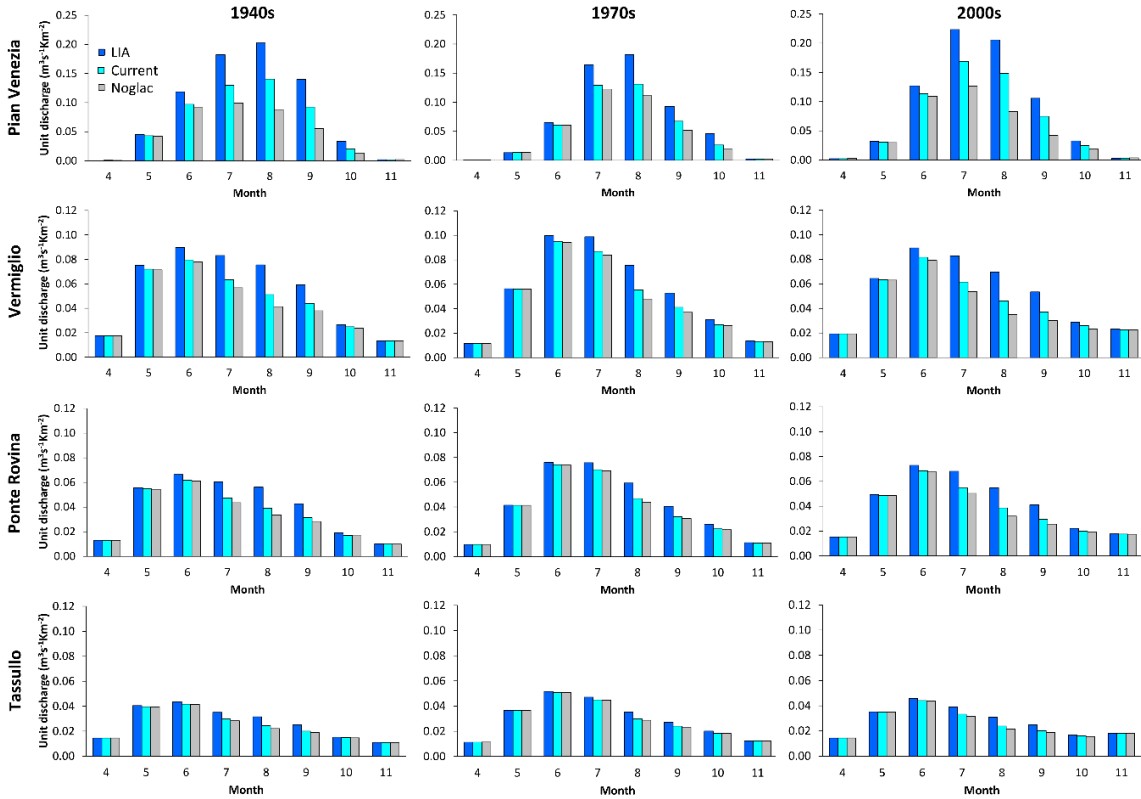

**Figure 5.** Monthly unit discharge calculated in the four catchments (rows) and in different meteorological periods (columns) with three different glacier extents (colors). Please, note that the vertical axis at Pian Venezia is different from the other three catchments.

In the larger catchments the peak in unit runoff is smaller and occurs earlier, in the first half of the ablation season. The larger the catchment, the earlier the peak. Changing the glacier extent in these large catchments does not lead to an anticipation of the summer peak, as observed for Pian Venezia. The most relevant change brought by the absence of glaciers is a more rapid recession after the late spring/early summer peak, and for the most glacierized catchments a significant decrease of this peak as well.

The seasonal variability of unit runoff shows a direct relationship with glacier extent, and an inverse relationship with catchment area, as shown by the flow duration curves and frequency distribution of unit discharge shown in Figure 6. Larger percent glacierization leads to higher spread and higher frequency of high runoff classes, and vice versa, in particular for the warm 1940s and 2000s. These effects are greatest at Pian Venezia and gradually decrease with catchment area, but are still discernible at Tassullo.

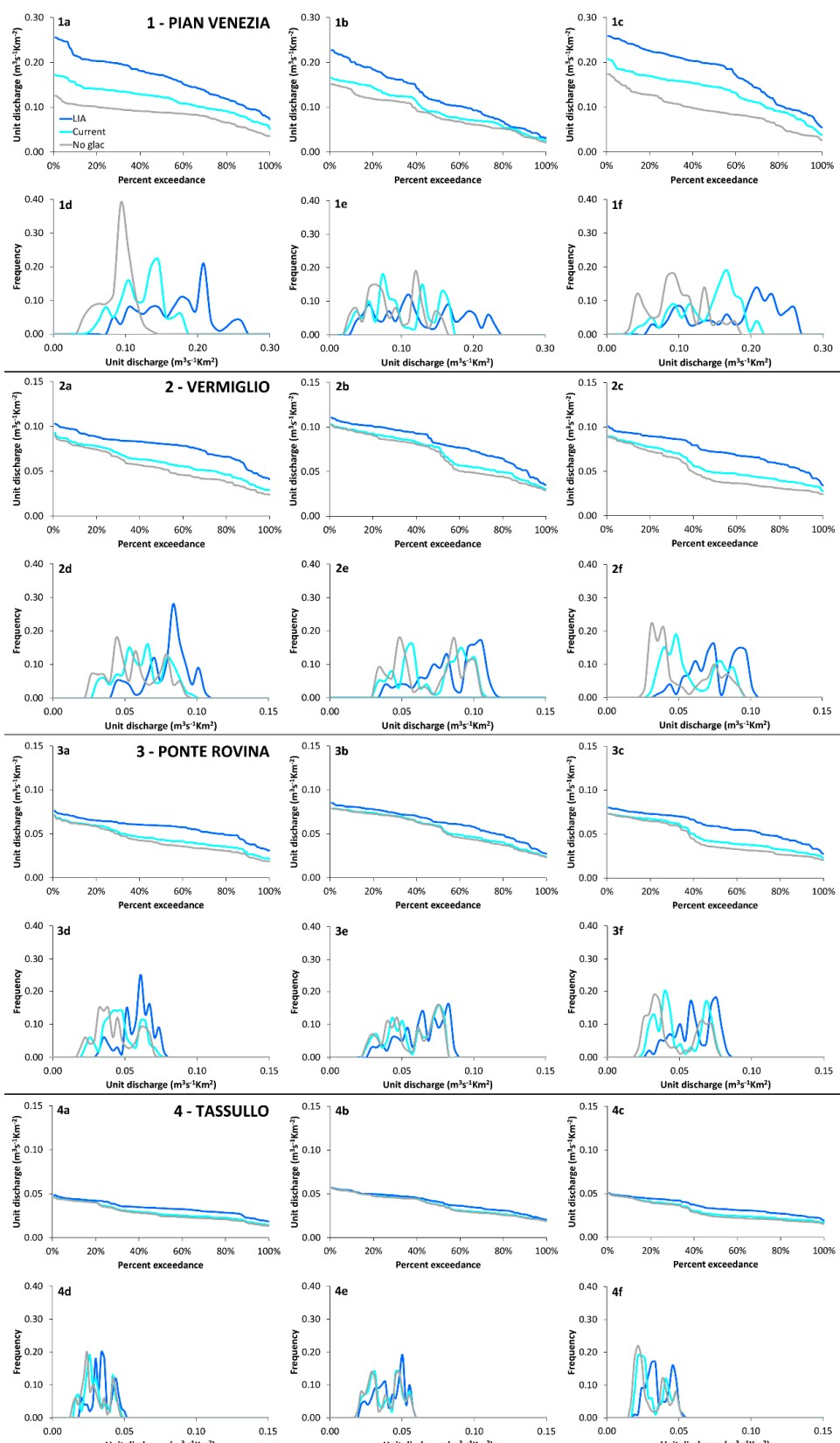

**Figure 6.** (**a**–**c**): 10-year averaged flow-duration curves of daily mean unit discharge from April to November in the four investigated catchments; (**d**–**f**): frequency distribution of the 10-year daily mean unit discharge. Please, note that the vertical axis at Pian Venezia is different from the other three catchments.

### 5.2. Scale Dependence of Hydrological Changes

With the Noglac conditions, the median unit discharge in August of 1940s is 30–60% smaller compared to the LIA glacier cover, varying among catchments, with differences that are directly proportional to initial glacierization and inversely proportional to the catchment area (Figure 7). Current glacier cover leads to 10–40% larger unit discharge compared to Noglac conditions, with same proportionality to initial glacierization and catchment area. Results for the 2000s meteorological period are very similar to the 1940s.

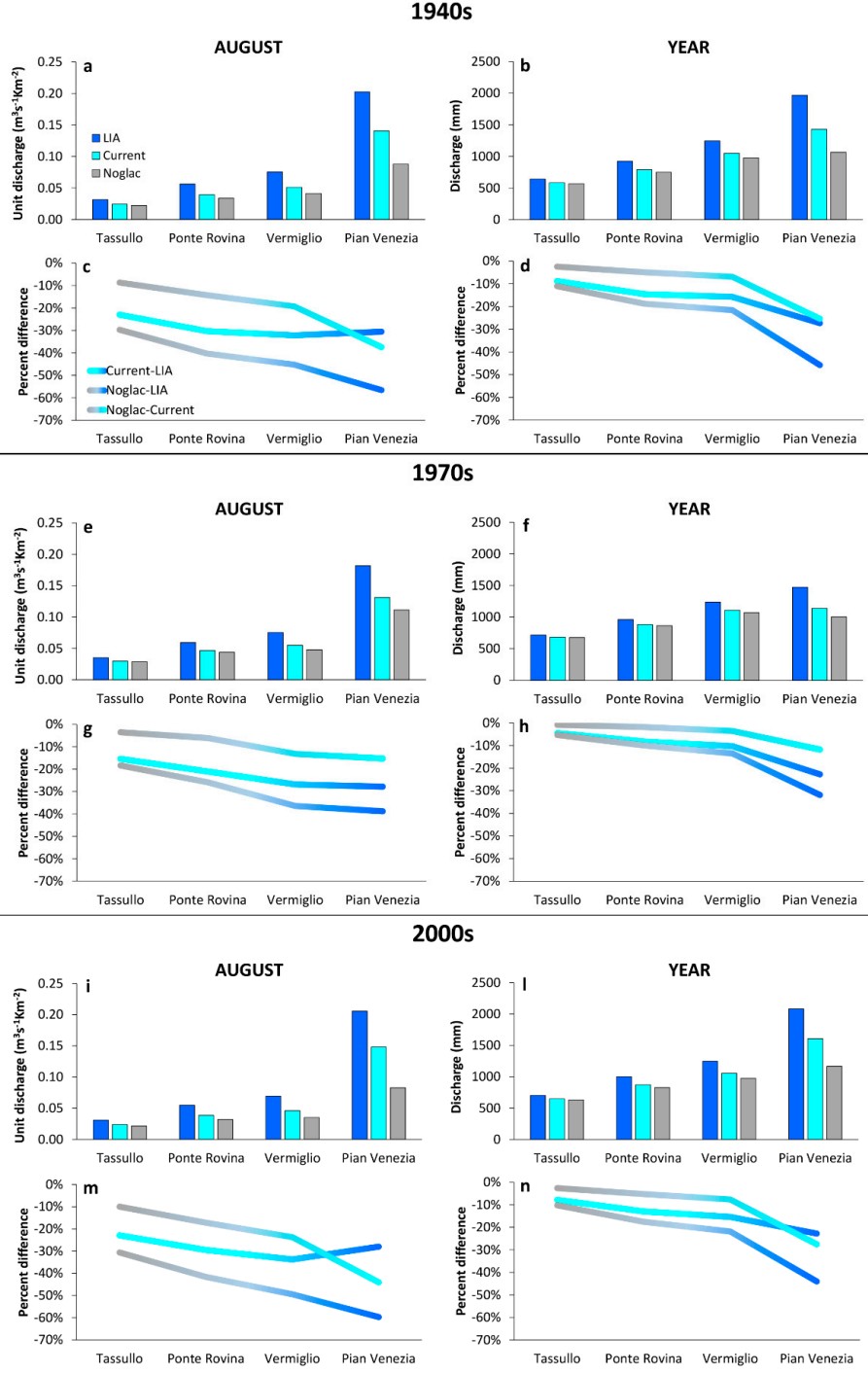

**Figure 7.** Comparison of median unit discharge in August (**left column**) and of annual runoff (**right column**) obtained with different glacier extents and in different meteorological periods.

The difference in August median unit discharge between the Current and the LIA glacier cover is more uniform, averaging 30% in all catchments. Considering the 10th percentile of August unit discharge (not shown), instead of the median, we found negligible increase of these differences. The smaller catchment, Pian Venezia, is the only one where the Current glacier cover, if compared with the Noglac conditions, ensures a unit runoff larger than its reduction calculated from the LIA (Figure 7c,m). In other words, the reduction in runoff from the LIA to the Current glacier conditions is smaller than the expected reduction after complete glacier disappearance.

Current glaciers still ensure higher August runoff in all climatic conditions, compared to the Noglac conditions. However, this 'damping' effect is largely decreased if compared to the LIA conditions, and this decrease is directly related to catchment area and initial glacierization.

Differences among glacier cover conditions are significantly smaller in the 1970s (Figure 7e,g). This is attributable to (i) decreased runoff from glacier melt, which is more effective over the larger LIA glaciers compared to Current glaciers, and (ii) to increased runoff from higher precipitation over bare ground in the Noglac scenario.

Similar results are obtained by analyzing the annual runoff, which obviously shows much smaller differences among glacier cover scenarios, with the exception of Pian Venezia. In this catchment, the decrease from LIA to the Current glacier cover is of the same magnitude of the expected reduction from Current conditions to complete glacier meltout. For the larger catchments, most of the glacier damping effect is already vanished, and there is very little difference between annual unit discharge obtained with Current glaciers or without them (Figure 7d,h,n).

*5.3. Hydrological Response under Extreme Meteorological Conditions*

5.3.1. The 2003 Hydrological Year

The LIA glacier cover would have ensured high runoff during the warm and dry 2003 summer season. In the three larger catchments, the August runoff would have been comparable to that of June and July, and at Pian Venezia nearly double that that (Figure 8), due to enhanced glacier melt.

In the Vermiglio catchment the LIA glacier extent would have led to an increase in daily runoff during the first half of August, whereas the Current and the Noglac conditions show a continuation of the decrease in runoff started in July. We can recognize this behavior also in the Ponte Rovina and Tassullo catchments, even if less evident.

Differences in unit discharge among glacier cover conditions are largest between July and August, but the period of significant water shortage with Current glaciers and Noglac spans across five months, from May to September. LIA glaciers, but also Current glaciers, lead to much higher runoff in mid- to late-summer, and this is observable in all catchments.

The difference between Current and Noglac conditions in 2003 is up to three times larger than with average conditions of 2000s (Figure 5). This increase is higher for the larger basins, while it is only 50% in the smaller basin where runoff is dominated by glacier melt in any case. This highlights the relevance of Current glaciers, which even though they have small residual impacts (compared to the LIA glaciers) on late summer runoff under 'normal' conditions, they are still important for mitigating excessive heat and drought. In conditions similar to August 2003, they ensure 30–60% higher mean runoff, and 50–80% higher minimum runoff, compared to deglacierized catchments. Even though the upper values in these ranges are obtained for the smaller catchments, with higher percent glacier cover, the difference is significant also for the larger catchments, in particular for minimum runoff.

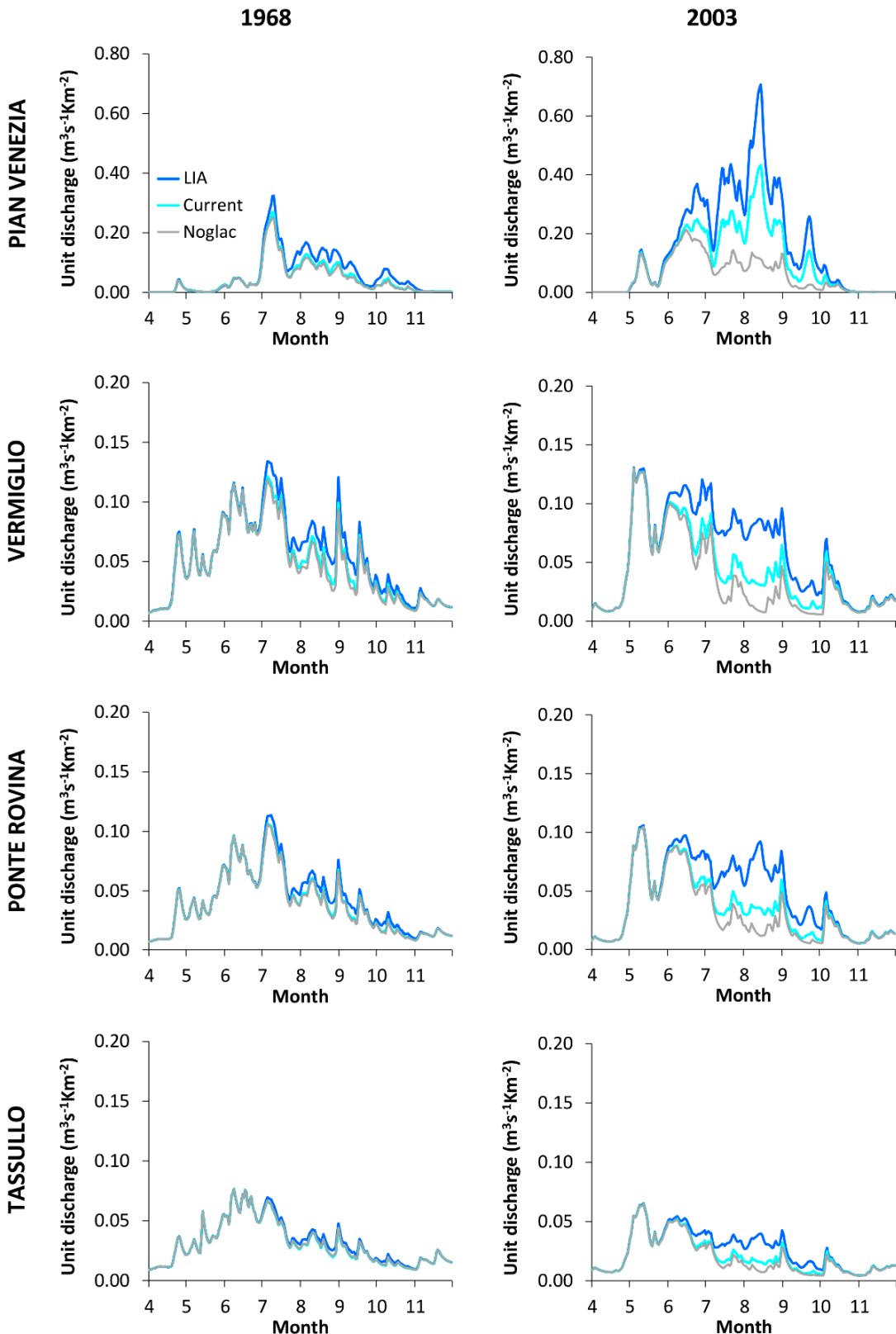

**Figure 8.** Daily unit discharge in 1968 (**left column**) and 2003 (**right column**) for different glacier extent (colors), in different catchments (rows). Please note that the vertical scale for Pian Venezia is different from the other three.

### 5.3.2. The 1968 Hydrological Year

Under cool and wet summer conditions, like in 1968, there is little if no difference between the runoff from Current and Noglac glacier cover (Figure 8). In the Vermiglio and Pian Venezia catchments, this difference hardly reaches 10% for average and 20% for minimum August runoff. Higher differences in runoff are obtained comparing the LIA and the Noglac extents, reaching 15–35% for mean August runoff and 20–45% for minimum August runoff. These differences decrease from smaller to larger catchments.

The difference among glacier extent conditions are almost zero until the end of June, and in the following months they are quite evenly distributed, without showing a clear peak in August as in 2003.

These results are due to the persistent snow cover at the glaciers' altitude during the summer 1968. Limited or null additional runoff has to be expected from the Current glaciers in these conditions, because they would have been covered by snow for most of the ablation season, and in most of their area. By the end of June, instead, the larger and lower-reaching LIA glaciers would have lost the snow cover in their ablation area, thus contributing additional runoff during the second half of the ablation season.

## 6. Discussion

### 6.1. Methodological Considerations

Assessing the sensitivity of the glacio-hydrological system of the study area based on past observations, instead of future projections, enabled to reduce the main source of uncertainty in sensitivity studies based on future climatic scenarios, whose prediction shows a considerable spread e.g., [13]. This was feasible thanks to the rare hydro-meteorological time series available in the study area.

Past climatic observations, however, are not free from problems. In particular, they are subject to inhomogeneities and measurement errors. The setup of a solid meteorological input was an essential prerequisite for obtaining meaningful results in our sensitivity analysis.

The homogenization technique adopted in this work was primarily intended to preserve as much as possible the original data. For this reason, we avoided using precipitation data from outside the study area to homogenize the raw series, taking in consideration the high spatial variability of precipitation in mountain areas [46]. In addition, we avoided correcting smaller inhomogeneities, in particular for temperature series, considering possible impacts in the solid vs. liquid partitioning of precipitation. Inhomogeneities in temperature measurements are indeed expected to be larger under stable weather conditions (high radiation, low wind speed), and smaller during precipitation events. Therefore, applying a correction to inhomogeneous periods in temperature series is expected to have a higher impact (excessive temperature correction) during precipitation events, further amplified by the application of precipitation correction factors.

Some assumptions have been made while pre-processing the meteorological data before modeling, mainly regarding the time/space invariance of precipitation correction factors and gradients of temperature and precipitation. Precipitation correction factors are likely different for different instruments and prevailing meteorological conditions during precipitation events. These differences should be higher at high altitude, were the fraction of solid precipitation and the wind speed are larger. Carturan [29] and Carturan et al. [47] investigated this problem at the high-altitude weather station of the Careser Diga (2605 m), finding negligible difference in precipitation correction factors between the old manual rain gauge, used until 1991, and the new tipping-bucket automatic heated rain gauge, in use since then. No information exist for other rain gauges of the study area, but given the smaller undercatch of precipitation at lower altitude, we think that the assumption of unchanged correction factors between the old manual instruments and the new automatic ones is reasonable.

Vertical gradients of precipitation and temperature exhibit a significant variability in space and time [48]. Monthly vertical gradients in temperature and precipitation were calculated in periods with

overlapping data from available weather stations located at different elevation: from 1940 to 1984 for precipitation, and from 2003 to 2014 for temperature. In absence of data from the same stations outside these periods, we have calculated a mean annual regime for temperature and precipitation vertical gradients, applying it unchanged in all the modeled time windows. This simplification, that does not consider the inter-annual variability of vertical gradients, was imposed by the availability of meteorological data.

The horizontal gradients of precipitation are accounted for using the spatial interpolation procedure described in Section 4.2. The horizontal variability of temperature, deriving for example from valley winds, warm and cold advection, local adiabatic heating, warming or cooling of different surfaces, is highly complex and difficult to model. The glacier cooling effect is a dominant factor of temperature variability, with relevant implications for glacier sensitivity to climatic changes over a long period of time [49,50]. However, multi-glacier fully-distributed modeling of the glacier cooling effect is not trivial and still in development [50–52]. For this reason, and because we have decided to focus on short time windows assuming unchanged glacier extent, the glacier cooling effect was not explicitly modeled, and was accounted for through calibration of melt parameters.

Depending on future climatic conditions, future glacier extent modeling is also subject to large uncertainty, further complicated by the existence of feedbacks that are still under investigation. Available reconstructions of past glacier extent enabled to investigate the hydrological response in our study area using constraints from actual measurements of glacier change.

While the Current extent and surface topography of glaciers are very well documented by recent LiDAR surveys, there are uncertainties in the reconstruction of glacier bedrock topographies (i.e., the Noglac scenario) and of surface topographies for the LIA glacier extent. The calculation procedure employed provided errors in single point ice thickness calculations of 20%, which is in line with previous works [32,33], and is acceptable considering the modeling simplifications.

One of the main issues of this work was finding information on glacier geometry as close as possible to the oldest hydro-meteorological data available (i.e., for the 1920s–1940s). We used the maximum Little Ice Age extent of glaciers, because reconstruction in this period is much more reliable, thanks to the well preserved and widespread landforms left by the glaciers [17], compared to more recent fluctuations in the late 19th Century or in the first half of the 20th Century. Assuming that in the 1920s–1940s the glacier extent was close to that of the LIA is allowable. Based on available reconstructions in the study area [53,54], the La Mare and Careser glaciers were only 9–10% smaller in the 1920s–1930s compared to the maximum LIA extent (see for example Figure 9). Reconstructions available for the Lobbia [55], Pisgana [56], and Solda glaciers [57], which are close to the study area, further support this assumption, as well as the comprehensive work of Desio [58]. The choice of focusing on single decades, keeping unchanged the glacier extent during these periods, was done because continuous modeling of glacier dynamics and geometric adjustment was outside the aims of this work.

The glacio-hydrological model that has been used in this work is a combination of a fully distributed module for the simulation of snow/glacier mass balance, coupled to a semi-distributed module for runoff routing. The choice of the modeling tools, and of the spatial domain to which they have been applied, was primarily dictated by the modeled processes. Because glacial processes typically show a high spatial variability, at the scale of tens of meters, they require sufficiently high spatial resolution to be taken into account with enough accuracy [59–61]. This is even more true for investigations on the sensitivity of snow and ice to climatic fluctuations, as in this case.

Hydrological processes also show high spatial variability, but conceptual and semi-distributed approaches are commonly used for modeling snow and ice hydrology in mountain catchments e.g., [62–65]. This is justifiable in this work with the fairly lower climatic sensitivity of the processes controlling runoff routing, compared to cryospheric processes. We had also to consider the availability of measurements to be used for constraining model parameters. In this work, the internal consistency of the snow/glacier mass balance module could be ensured by detailed measurements of snow and ice

mass balance, at various spatial scales, whereas runoff routing parameters could be constrained using lumped data, i.e., outlet streamflow measurements.

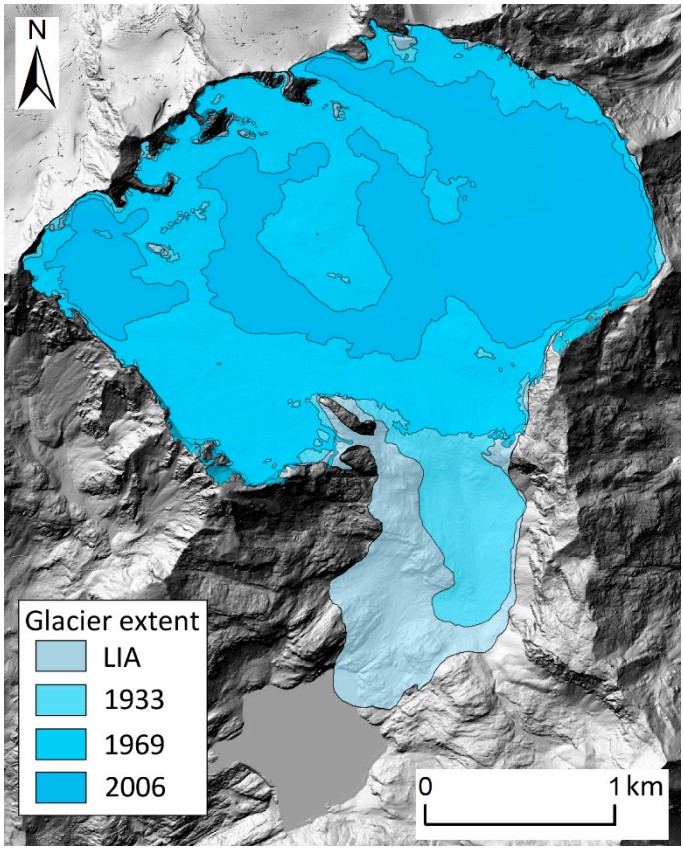

**Figure 9.** Careser Glacier extent in various periods since the LIA maximum [17,22,23].

*6.2. Considerations on the Sensitivity Analysis Results*

The climatic conditions in the 1940s were rather similar to those in the 2000s, in particular for the warm summer temperature and for the seasonal distribution and amount of precipitation. The heavy precipitation in November 2002, November and December 2008 and November 2010 make the 2000s more snowy, on average, at the glaciers' altitude. The 1970s were significantly more favorable for glaciers (leading to their observed expansion between the 1970s and 1980s, [66]). Temperature was ~2 °C lower than in the 1940s and 2000s between March and September, and precipitation was higher, in particular during winter and spring. The solid fraction was also larger, principally in summer, with important effects on the summer balance of glaciers [53,67].

The similarity between the 1940s and the 2000s is remarkable, as well as the particular conditions measured in the 1970s. This behavior is likely affected by multi-decadal patterns of oceanic thermal anomalies, like the Atlantic multidecadal oscillation (AMO) and the Pacific decadal oscillation (PDO), further modulated by continental-scale circulation patterns such as the North Atlantic oscillation (NAO) and the East Atlantic pattern (EA), which influence the temperature and precipitation distribution in winter and summer across Europe [68–70]. Most of these patterns favored positive glacier mass balance (i.e., positive storage) in the 1970s, and negative mass balance (i.e., negative storage) in the 1940s and 2000s.

These analogies/differences among the three periods are clearly visible in the hydrological response of the four analyzed catchments (Figures 5–7). The 2000s and 1940s behave very similarly, and with distinct differences when compared to the 1970s. As found e.g., by Huss, [5], smaller catchments with higher glacierized area fraction are comparatively more affected by changes in

glacier storage, while larger catchments are more affected by changes in the precipitation regime and seasonal snow accumulation/ablation. Climatic conditions during the 1970s are emblematic. There is significantly higher runoff in mid-summer during the 1970s in the larger catchments, and differences from the 1940s and 2000s increase from LIA to Current to the absence of glaciers (Figure 7). Therefore, these differences are not attributable to enhanced glacier runoff, but to specific climatic characteristics of the 1970s. In particular, the larger snow accumulation from October to May ensures sustained runoff from seasonal snow during summer, also in case of absence of glaciers. In the smaller catchment there is an opposite behavior, with higher mid/late summer runoff due to glacier melt in the 1940s and 2000s, and lower runoff in the 1970s due to positive glacier storage.

A considerable change in the hydrological regime from June to October was detected in response to decreasing glacier cover. This change consists mainly in a strong decrease of unit discharge after the seasonal snow is melt, in the second half of summer, with a maximum decrease in the month of August (Figure 5). There is also the tendency to an anticipation of the runoff peak from August to July in the smallest and most glacierized basin, whereas this effect is not visible on larger catchments where seasonal snow dominates the warm season runoff regime. Under the 2000s meteorological conditions, the peak is in July also for the LIA glacier extent, due to the warmer spring temperature compared to the 1940s and 1970s, in combination to higher snow accumulation outside the glaciers in autumn and early winter. Variations in summer runoff among glacier cover conditions are much more evident for the warm 1940s and 2000s meteorological periods, compared to the cooler 1970s.

These results highlight increased hydrological sensitivity towards warmer climatic conditions, reinforced by glacier decay, and suggest impacts for downstream catchments, in spite of their very small percent glacierization. According to the literature, macroscale transboundary catchments, as for example the Po valley, benefit from glacier runoff during the second half of summer in a significant way. Even if they are far from glaciers, the runoff from glacier melt is relevant because in the plains the summer is much warmer and drier, with strong evapotranspiration. Huss, [5], for example, demonstrated that glacier contribution to August runoff at Pontelagoscuro and Piacenza reaches 15–20% on average, with peaks of 30% during warm and dry summers, like in 2003.

In the 2000s, the glacier contribution in August (i.e., the negative glaciers storage change from the water balance equation) ranges from 43% to 84% with the LIA glacier cover, and from 26% to 78% with the Current glacier cover, with percent contributions that are inversely proportional to catchment area (Figure 10). The decrease in glacier contribution with increasing catchment area is not linear. The largest decrease is observed between the smaller catchments, in our case between the Pian Venezia and Vermiglio basins, and becomes much slower for larger catchments. We do not observe the increasing importance of glacierization, described for macroscale catchments by Huss [5], because our larger catchment (Tassullo) is rather small in comparison, and because in the study area the climatic conditions are rather homogeneous.

Glacier contribution can strikingly increase during extreme conditions, like in summer 2003. Figure 10 shows that in August 2003 the percent glacier contribution is nearly twofold for the three larger catchments analyzed, compared to average meteorological conditions in the 2000s, reaching 58% at Tassullo. The increase in the headwater catchment is of only 15% because glacier runoff already dominates under average meteorological conditions. Based on these results, relevant impacts from glacier vanishing are expected during warm and dry years like 2003, not only in headwater catchments but also on downstream catchments. Similar results were obtained for example for the Skykomish River, Washington, by Pelto [10].

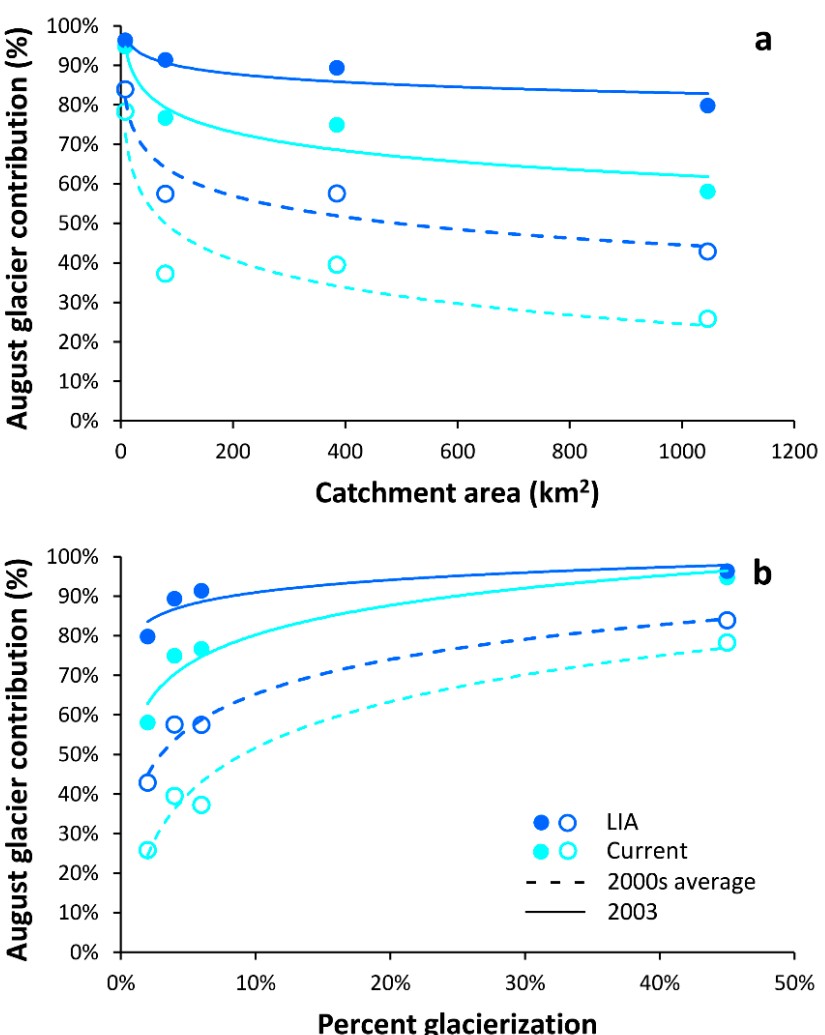

**Figure 10.** Percent glacier contribution to August runoff vs. (**a**) catchment area, and (**b**) percent glacierization. Average contribution in the 2000s (decade from 2003 to 2012) is compared with contribution in 2003. Tentative logarithmic regressions lines are applied to each series.

When compared to glacio-hydrological experiments that use future modeled atmospheric warming scenarios [71,72], these results confirm a strong decrease in the capability of glacierized catchments in damping the runoff fluctuations caused by precipitation/temperature variability. This decrease is related to the progressive melt out of residual ice masses, but is preceded by a period of increased runoff derived from negative glacier storage change, with a peak that is expected between 2020–2040 for highly glacierized catchments [7,73,74]. It is therefore interesting to assess 'at which point' our study area is in this transition. To do that, we have quantified the residual damping effect (RDE) of Current glaciers, compared to that of the LIA glacier extent

$$RDE = \frac{DE_{2000s}}{DE_{LIA}} \cdot 100 \tag{3}$$

where *DE* is the damping effect, calculated as

$$DE = q_{gc} - q_{ga} \tag{4}$$

where, $q_{gc}$ is the unit discharge of August with the LIA or Current glacier cover, and $q_{ga}$ is the unit discharge of August in absence of glacier cover. Results are displayed in Table 5, and show that in

the smaller and most glacierized catchment RDE is around 50%, whereas in the larger catchments it is about 30% (24% for the Tassullo catchment with the 2003 meteorological conditions). This means that three/quarter of the LIA damping effect are already vanished from the study area, and only in headwater catchments it still reaches half of the original value.

**Table 5.** Unit discharge ($m^3s^{-1}$ $km^{-2}$) in August during the decade from 2003 to 2012, and in 2003. Value in bold indicate 2003 unit discharge larger than average 2003–2012 unit discharge. Residual damping effect (RDE) for the Current glaciers are shown in brackets.

| Catchments | LIA Glacier Extent | | Current Glacier Extent | | No Glaciers | |
|---|---|---|---|---|---|---|
| | 2003–2012 | 2003 | 2003–2012 (RDE) | 2003 (RDE) | 2003–2012 | 2003 |
| Pian Venezia | 0.206 | **0.448** | 0.148 (53%) | **0.282** (53%) | 0.083 | **0.098** |
| Vermiglio | 0.069 | **0.080** | 0.046 (32%) | 0.037 (32%) | 0.035 | 0.017 |
| Ponte Rovina | 0.055 | **0.073** | 0.039 (30%) | 0.035 (30%) | 0.032 | 0.019 |
| Tassullo | 0.031 | **0.033** | 0.024 (30%) | 0.017 (24%) | 0.021 | 0.012 |

The summer of 2003 has been frequently referred to as a possible example of future climatic conditions during summer in the Alps, under sustained $CO_2$ emissions scenarios [75,76]. To reveal if the peak in runoff due to glacier wastage still has to come, or has already passed in the study area, we compared the runoff in August 2003 with the runoff under 'average' conditions during the 2000s, with different glacier cover. Results show that the LIA glaciers would have ensured higher unit runoff on all four analyzed basins in August 2003, compared to average 2000s meteorological conditions (Table 5). In contrast, the Current glacier cover enables increased unit runoff only in the Pian Venezia basin.

These results are not conclusive because a complete assessment would have required continuous modeling of climate, glacier mass balance, glacier dynamics, geometric adjustments, and runoff for a sufficiently long time period, spanning the latest decades and the next two or three decades. However, the values displayed in Table 5 strongly suggest that the peak in late-summer runoff under warming climate, due to glacier wastage, has already passed in Val di Sole, with the exception of few glacierized headwater catchments. Because more than half of the glacier area has been lost from most of the watersheds (Table 1), it would require more than a doubling in summer ablation compared to the 2003–2012 mean to be offset, which has a low probability of occurrence.

Koboltschnig and Schöner, [9], came to similar results, finding that only Austrian catchments with glacierization larger than 10% were able to provide larger runoff in August 2003, compared to mean long-term August runoff. Other works in the recent literature agree with our findings and indicate that the peak in runoff under warming scenarios is expected earlier for catchments with lower initial glacierization and smaller/thinner glaciers [7,74,77], or has already passed [10,78–80].

It is interesting to note that, also without glaciers, there is a small increase in unit runoff during August 2003 at Pian Venezia, compared to 2000s average meteorological conditions. This increase is brought by the melt of snow accumulated at high altitude in previous years (impending formation of small glaciers during model initialization). However, this marginal increase in unit runoff and the formation of small glaciers is observed when 2003 represents an exceptionally warm summer, and are unlikely to occur when 2003 is assumed to represent average meteorological conditions.

## 7. Conclusions

In this study, we have analyzed the hydrological sensitivity of four glacierized catchments located in the Eastern Italian Alps, with different area and percent glacierization, under different climatic conditions and with three glacier extent conditions. Investigations were carried out using a glacio-hydrological model and were based on past glacier change and hydro-meteorological series, rather than using future projections of climate and glacier dynamics. This approach had the advantage

of reducing uncertainty on future climatic conditions and glacier response, and of ensuring high internal consistency in glacio-hydrological modeling.

A possible drawback of this method lies in neglecting possible extreme future conditions, outside the range of variability given by observations in the 20th and 21st centuries. To partly compensate for this, we investigated the hydrological response of the catchments with a complete absence of glaciers (calculating the shape of the bedrock underneath Current glaciers), and in recent extreme years, like 2003.

The study reveals that a progressive transition from a glacial regime to a nival regime is under way in the analyzed catchments, caused by the shrinking of glaciers in response to warming climate. With smaller glaciers (as mapped in 2006) and warmer meteorological conditions (average conditions from 2003 to 2012), the runoff is strongly decreased after the seasonal snow has melted, in the second half of summer, compared to results obtained using the LIA glacier scenario. The runoff peak tends to shift from mid- to early summer, and this tendency is more evident in the headwater catchments.

Different glacier cover scenarios (LIA, Current and absence of glaciers) present the highest difference in hydrological response in the month of August, and effects are more evident during periods of glacier wastage as in the 1940s and in the 2000s, and in the smaller catchments with high percent glacierization. These results highlight increased hydrological sensitivity towards warmer climatic conditions, reinforced by glacier decay.

Compared to the absence of glaciers, Current glaciers still ensure higher runoff during summer, in all climatic conditions and basins considered, with significant contribution to late summer runoff also in the larger catchments. However, this glacier damping effect, i.e., the difference between late summer runoff with or without glacier cover, is largely decreased if compared to the LIA conditions. This decrease is larger for wider basins. If smaller and highly glacierized catchments still preserve ~50% of the initial damping effect in August, the larger catchments keep only 25–30% of it.

However, this effect is still relevant in extremely warm and dry conditions (like in 2003), when Current glacier contribution to late summer runoff in the larger catchment reaches ~60%, and it does not scale linearly with catchment area and percent glacierization, in agreement with previous works.

In the study area, only the LIA glacier cover would have ensured increased runoff in 2003, compared to 2000s (2003 to 2012) average meteorological conditions, resulting from enhanced glacier melt. Current glaciers are no longer able to compensate for warm/dry conditions, with the exception of small headwater catchments. This suggests that the expected peak in runoff under warming climate, attributable to glacier melt, has already passed in this area.

Future conditions of nearly complete deglacierization, combined with summer conditions similar to 2003, would have strong impacts in the runoff of the study area. Without glacier cover, we obtained a small increase in 2003 runoff only in the smallest catchment, compared to 2000s average conditions. This increase is brought by the melt of snow accumulated at high altitude in previous years, but is only possible when 2003 represents an exceptionally warm summer, whereas it is unlikely to occur if 2003 represents 'normal' future climatic conditions.

**Author Contributions:** Conceptualization, L.C., F.C., M.B., and G.D.F.; Methodology, L.C., F.D.B., F.C., G.D.F., M.B., and D.Z.; Software, F.C. and P.B.; Validation, F.D.B.; Formal analysis, F.D.B. and L.C.; Data curation, F.D.B.; Writing—original draft preparation, L.C. and F.D.B.; Writing—review and editing, L.C., F.D.B., F.C., D.Z., P.B., M.B., and G.D.F.; Visualization, L.C. and F.D.B.; Supervision, G.D.F.; Funding acquisition, G.D.F. and M.B.

**Funding:** This study was carried out in the framework of a PhD studentship at the Department of Land, Environment, Agriculture and Forestry, funded by the University of Padova. The study was also funded by (i) the Italian MIUR project (PRIN 2010–2011) "Response of Morphoclimatic System Dynamics to Global Changes and Related Geomorphological Hazards" (local and national coordinators G. Dalla Fontana and C. Baroni); (ii) the University of Padova (Progetto Giovani—Bando 2013—Senior Research Grant: 'Impact of climatic fluctuations on snow- and ice-dominated alpine watersheds: effects on the cryosphere and hydrology'); (iii) the BORG_EPPR_P01_01 fund; and (iv) the HYDRATE Project (cooordinator Marco Borga).

**Acknowledgments:** The authors would like to thank the numerous friends, colleagues, alpine guides, and students who contributed to the collection of the field data. The Autonomous Province of Trento is acknowledged for providing the hydro-meteorological and topographic data. Many thanks to Carlo Baroni

and Roberto Dinale for their useful suggestions. The authors are also grateful to Mattia Callegari and Claudia Notarnicola (EURAC research—Institute for Applied Remote Sensing) for providing the MODIS-derived snow cover maps.

**Conflicts of Interest:** The authors declare no conflict of interest.

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
