# Peer review of "Relevance and Scale Dependence of Hydrological Changes in Glacierized Catchments: Insights from Historical Data Series in the Eastern Italian Alps"

_water, doi:10.3390/w11010089_

Round 1
Reviewer 1 Report
Carturan et al (2018) provide a detailed examination of glacier runoff changes from four headwaters basins in Italy. The authors use long term data sets to help validate the model used. The authors can expand their referencing of supporting studies outside the region. It is worth sharing more detail about glacier mapping and specific mass balance measurements. In the introduction or conclusion the hydropower developments on the Noce must be briefly described, such as Cogolo and Malga Mare [56]. With further supporting data this will be a valuable contribution.
33: Need a more recent reference to compliment the earlier references Zemp et al (2015) is a possibility.
49: The timing of peak glacier runoff has been examined in detail by a number of authors. Bliss et al (2014; Fig. 4) and Huss et al (2017: Fig 4) identify that many regions have already had the peak. The statement that is expected in the next two to three decades is not applicable overall.
51: Worth providing a summary of the offsetting mechanisms for glacier runoff change.
The amount of glacier runoff is the product of surface area and ablation rate. Glacier volume loss can contribute to changes in streamflow, leading to an increase in overall streamflow if the rate of volume loss is sufficiently large, or a decline in streamflow if the area of glacier cover declines sufficiently to offset any increase in ablation rate (Pelto, 2011).
63:”… hydro-meteorological data from the Noce River Basin in the Italian Alps. Some of these records are amongst the longest…”
85: Remove “wide”
95: “..with area ranging from 8.4 to 1046 km2 and percent glacierization from 2 to 45%.”
97: What is reference for the Table 1 data if not from this study?
104: Remove “comprised”
112: Change “spotting out” to “identifying”
131: Given rate of change 2006 represents a significantly larger glaciated area than now.
181: Table 3 is highly redundant and can be reduced significantly.
260: I suggest you consider adding a section in 3.1 on the LeMare Glacier observations; this will increase the understanding of how strong the validation/calibration is.
280: At some place it would be useful to provide a sample map of glacier change such as Figure 6 in Carturan et al (2013), with four different moments LIA, 1940, 1970, now.
310 and 313: The percentage range of the runoff change of 30-60% and 10-40% is large. Is this because of the variation between basins? Is it better at the basin level?
328: Why is catchment area so critical?
446: This assumption of similarity needs better corroboration. Give examples of % change from actual observation in Italian or Swiss Alps. Carturan et al (2013) noted that “the first topographic survey of the Careser glacier was carried out in 1933 using terrestrial photogrammetry techniques with subsequent surveys in 1959, 1969, 1980, 1990 and 2000. Does Careser fit the proposed assumption?
531: This has been observed in the Skykomish River basin as well, indicating it is a relationship not exclusive to the Alps (Pelto, 2011).
536: The references for the peak flow are not the most recent. More recent studies indicate the peak has been reached in the study area.
570: In fact there are basins that have been documented to have already passed peak flow (Stahl and Moore, 2006; Pelto, 2011; Baraer et al 2012).
613: I agree with the following conclusion; however, this conclusion should be noted specifically near 531 or 536. “ This suggests that the expected peak in runoff under warming climate, 613 attributable to glacier melt, has already passed in this area.”
1. Bliss, A., Hock, R., and Radic, V. Global response of glacier runoff to twenty-first century climate change, J. Geophys. Res. Earth 2014, 119, 717–730. doi.org/10.1002/2013JF002931.
2. Baraer, M., B. G. Mark, J. M. McKenzie, T. Condom, et al. Glacier recession and water resources in Peru’s Cordillera Blanca. J. Glaciol., 2012, 58, 134–150. https://doi.org/10.3189/2012jog11j186.
3. Huss M. et al. Toward mountains without permanent snow and ice. Earth’s Future 2017, 5, 418–435, doi:10.1002/2016EF000514.
4. Pelto, M.S. Skykomish River, Washington: Impact of ongoing glacier retreat on streamflow. Hydrological Processes 2011, 25, 3267–3371.
5. Stahl, K. and Moore, D. Influence of watershed glacier coverage on summer streamflow in British Columbia, Canada. Water Resour. Res. 2006, 42, W06201, doi: 10.1029/2006WR005022.
6. Zemp, M.; Frey, H.; Gärtner-Roer, I.; Nussbaumer, S.U.; Hoelzle, M.; Paul, F.; Haeberli, W.; et al. Historically unprecedented global glacier decline in the early 21st century. J. Glaciol. 2015, 61, 745–762.
Reviewer 2 Report
The MS by Carturan et al. presents a nice interesting scientific story about hydrological response in 4 catchments with varying glacier cover using ground observations and modelling. The MS is well written except for minor language issues which at times impede the flow. There are some minor scientific concerns, appended below, that need to addressed:
Line 15: Sensitivity analysis pertaining to what!! Needs to be mentioned here
Line 15: Superscript "km2". Mention percentages within brackets adjacent to areas
Line 20: "has" instead of "is"
Line 32: Cite
Line 35: Replace "brings" with a better word here
Line 43: Glacierization appears like a process. Why cant the authors mention percent glacier cover!
Line 55: "Run" or "simulate"
Line 63: Should be "time series"
Line 78: Cite the source of geological formations
On Study Area map: How have the four catchments been delineated! It appears that it is a single watershed and hydrological considerations for sub-catchment delineation have been bypassed
Line 87: What is the source of this information?
Line 87: "outside" or "adjacent"
Line 96: In the abstract authors say that the percent glacier cover ranges between 2 and 70%. Isn't this contradictory!
Line 99 on table: Source of this info!
Line 114: Use "after homogenization"
Line 116: "precipitation data"
Line 118: "hydrometric" or "hydrometeorological"
Line 132: Why was DEM degraded (resampled to 30m)?
Line 208: The lapse rates are highly variable depending upon the adiabatic conditions,altitude range and seasonality (See Romshoo et al. 2018, Journal of Mountain Science, https://link.springer.com/article/10.1007/s11629-017-4566-x). Why have authors used just two lapse rates! How was the threshold elevation computed!
Line 221: Why was IDW used compared to other methods (geographically weighted interpolation, krigging or natural neighbour)
Line 249: What is the source of land use data?
Line 244: It would have been prudent to use 4 areas: 1-above vegetated landscape as one zone, 2-the pastures/shrubland areas between 2200-2700, 3-1000-2200 m comprising of conifers and, 4: below 1000. The authors need to defend their definition of elevation bands and clarify
Line 265: "Nice" or "robust"!! Quantify
Line 290: Although there is a significant change across scenarios but I am intrigued that there is no significant shift in the runoff. Something like more runoff in late Spring for 1970s and 2000s as compared to Summer esp when authors say that 'peak runoff' from glacial melt has already passed. Why is it so! (See Romshoo and Marai 2018, J of Mountain Science, https://link.springer.com/article/10.1007/s11629-017-4474-0)
Line 356: Reframe and correct the language
Line 392: Although cited in text, but the authors need to explain the actual homogenization of meteorological data especially those associated with instrumentation change
Line 429: Line 392-430 could be moved to results
Round 2
Reviewer 1 Report
This paper as presently written provides a quantitative assessment of runoff changes due to climate change and glacier change in specific Italian Alpine watersheds. The following four points are all minor and do not require additional review.
96: Please add a bit of detail on the hydropower plants, are they run of river or reservoir based, what is their power output?
591: It is worth reiterating that given more than half of the glacier area has been lost from most of the watersheds in Table 1, it would require more than a doubling in summer ablation to offset. The general principal was stated at Line 51, now apply specifically here. A doubling of ablation rate is not realistic and again indicates peak water has occurred.
Fig 4. Put month names instead of numbers on x-axis
Fig 6. I am concerned about losing quantitative value by including 24 graphs in one. Can this number be reduced so that each sub-graph illustrated can be expanded?
Author Response
Dear Editor,
we thank the Reviewer 1 for its careful reading and final notes. We report here the replies to his comments:
Line 96: Please add a bit of detail on the hydropower plants, are they run of river or reservoir based, what is their power output?
Reply: ok, information added
Line 591: It is worth reiterating that given more than half of the glacier
area has been lost from most of the watersheds in Table 1, it would
require more than a doubling in summer ablation to offset. The general
principal was stated at Line 51, now apply specifically here. A doubling
of ablation rate is not realistic and again indicates peak water has
occurred.
Reply: ok, sentence added as suggested
Fig 4. Put month names instead of numbers on x-axis
Reply: ok, figure modified as suggested
Fig 6. I am concerned about losing quantitative value by including 24 graphs in one. Can this number be reduced so that each sub-graph illustrated can be expanded?
Reply: we had to put all these graphs in this figure because they represent all the combinations tested. In our opinion they are all necessary for the readed to understand the results of the analysis and to follow their description in the text.